# Si-C is a method for inferring super-resolution intact genome structure from single-cell Hi-C data

Luming Meng [1✉], Chenxi Wang[2], Yi Shi[3] & Qiong Luo[2]

There is a strong demand for methods that can efficiently reconstruct valid super-resolution intact genome 3D structures from sparse and noise single-cell Hi-C data. Here, we develop Single-Cell Chromosome Conformation Calculator (Si-C) within the Bayesian theory framework and apply this approach to reconstruct intact genome 3D structures from single-cell Hi-C data of eight G1-phase haploid mouse ES cells. The inferred 100-kb and 10-kb structures consistently reproduce the known conserved features of chromatin organization revealed by independent imaging experiments. The analysis of the 10-kb resolution 3D structures reveals cell-to-cell varying domain structures in individual cells and hyperfine structures in domains, such as loops. An average of 0.2 contact reads per divided bin is sufficient for Si-C to obtain reliable structures. The valid super-resolution structures constructed by Si-C demonstrate the potential for visualizing and investigating interactions between all chromatin loci at the genome scale in individual cells.

[1] MOE Key Laboratory of Laser Life Science and Guangdong Provincial Key Laboratory of Laser Life Science, College of Biophotonics, South China Normal University, Guangzhou, China. [2] Center for Computational Quantum Chemistry, School of Chemistry, South China Normal University, Guangzhou, China. [3] Bio-X Institutes, Key Laboratory for the Genetics of Developmental and Neuropsychiatric Disorders, Shanghai Jiao Tong University, Shanghai, China. ✉email: menglum@scnu.edu.cn

Chromatin folds into an intricate three-dimensional (3D) structure to regulate cellular processes including gene expression[1–4]. Recent researches on enhancer perturbation[5,6] revealed that many gene regulations are associated with higher-order interactions that involve three or more chromatin loci. A new opinion appears to be becoming popular that enhancer–promoter pairing, both in the sense of the 3D genome and function, is not binary but quantitative[3,7]. The currently reported experiment of super-resolution chromatin tracing confirmed that cooperative higher-order chromatin interactions are widespread in single cells and, moreover, such interactions exhibit substantial cell-to-cell variation[8]. In addition, in mammalian genomes, most enhancers are tens to hundreds of kilobases away from their target promoters[6]. These findings highlight the critical need for super-resolution 3D genome structures of individual cells.

Owing to advances in genome-wide chromosome conformation capture technology (Hi-C) and decreasing sequencing costs, single-cell Hi-C protocols have been available[9–12]. Computational modeling of genome structure from single-cell Hi-C contact data provides a pivotal avenue to address the critical need[9,13–17]. However, only a small portion of contacts between genomic loci can be probed in a single-cell Hi-C experiment[18]. The extreme sparsity and noisy of contacted information thus pose a huge challenge for determining biologically valid and high-resolution 3D genome structure from single-cell Hi-C data.

A class of methods based on multidimensional scaling (MDS) was developed to infer 3D structure from single-cell Hi-C data. Shortest-path reconstruction in 3D (ShRec3D)[13] models 3D structure of a chromosome through applying MDS to a constructed Euclidean distance matrix consisting of pairwise distances between different loci in the chromosome. In the distance matrix, ShRec3D defines a contact distance between the locus pair that show a contact in the Hi-C data and a similar distance between adjacent loci along the chromosome sequence. After that, the missing distances in the matrix are imputed by the shortest-path distance that is derived from a graph constructed based on the Hi-C data and the genome connectivity. Paulsen et al. suggested that the importance and accuracy of various imputed distances should be distinguished with weights and developed manifold-based optimization (MBO)[15] method to address the concern. Comparing to ShRec3D, MBO allows for flexibility in the determination of 3D chromosome structure from single-cell Hi-C data. Another class of popular methods for reconstructing 3D structure from single-cell Hi-C data is known as constraints optimization methods[9,14,16,19,20]. These methods describe chromatin fiber as a polymer consisting of beads of the same size and introduce a cost function taking into account input Hi-C data and polymer-physics properties as constraints. Then, an optimization or sampling procedure is performed to minimize the cost function and finally return a structure that satisfies the predefined constraints. For instance, Zhu et al. developed single-cell lattice (SCL) approach[19], whose cost function includes three terms, two of which are associated with the bead pairs that have a contact and the pairs without a contact (and simultaneously far away from the bead pairs that have a contact). The third one is a 2D Gaussian function to model the bead pairs that have no Hi-C contact, but sequentially close to the bead pairs with a contact. SCL uses Metropolis–Hastings simulation and simulated annealing to obtain 3D structures of the X chromosome of a mouse TH1 cell and chromosome 11 of a mouse ES cell and so on. Stevens et al. developed NucDynamics[9], which uses a force field with two terms: one representing a general repulsion between all sequentially nonadjacent beads in the chromosome and the other describing the distance restraint between the bead pairs that show a contact in Hi-C data or that are adjacent along

the chromosome sequence. Combining simulated annealing with molecular dynamics, NucDynamics successfully reconstructs intact genome 3D structures of a mouse ES cell from the single-cell Hi-C data for the first time.

In principle, MDS-based methods and constraints optimization methods implicitly assume that there is a unique 3D structure underlying the single-cell Hi-C data. However, different conformations can satisfy the incomplete Hi-C data and cannot be distinguished by experiments. To overcome the problem, constraints optimization methods repeat the optimization procedure multiple times from different random initial structures to generate structure ensembles. Although such a practice seems plausible, the variability of the structure ensemble cannot reflect the valid structure error bar, since the ensemble lacks a statistical foundation.

To address the issue, Carstens et al.[21] adopted the Bayesian inferential structure determination (ISD)[17] originally developed for reconstructing structures of proteins from the nuclear magnetic resonance data. The Carstens et al. method[21] builds a posterior probability distribution combining experimental single-cell data with polymer-physics-based model of chromatin fiber. Through using Markov chain Monte Carlo algorithm, a sampling procedure is carried out to produce a statistically meaningful structure ensemble that represents the posterior probability. The probabilistic nature of the Carstens et al. method makes it efficient for structure determination from sparse and noisy data. However, this method is resource-consuming and thus unsuitable for large-scale genome structure determination. Therefore, developing a Bayesian statistical method with the ability to reconstruct entire genome 3D structures at high resolution from single-cell Hi-C data of mammalian cells is desired.

Here, we develop single-cell chromosome conformation calculator (Si-C) within the Bayesian theory framework. Briefly, Si-C resolves a statistical inference problem with a fully data-driven modeling process (for the schematic illustration of Si-C method see Fig. 1). The goal of Si-C is to obtain the 3D genome conformation ($\mathbf{R}$) with the highest probability under the given Hi-C contact constraints ($\mathbf{C}$) being satisfied. We define the conditional probability distribution $P(\mathbf{R}|\mathbf{C})$ over the whole conformation space with Bayesian theorem. After that, the total potential energy of a 3D conformation is defined as $E(\mathbf{R}) = E_{\text{count}}(\mathbf{R}) + E_{\text{phys}}(\mathbf{R})$, where $E_{\text{count}} = -\ln P(\mathbf{R}|\mathbf{C})$ describes the potential energy derived from contact constraints and $E_{\text{phys}}(\mathbf{R})$ is a harmonic oscillator potential ensuring the connection between consecutive beads, and then an optimization procedure is performed to minimize $E(\mathbf{R})$, or in other words, to maximize $P(\mathbf{R}|\mathbf{C})$. To rapidly achieve the optimized structure at target resolution, we incorporate a hierarchical optimization strategy into Si-C framework. For each cell, the whole calculation of a given resolution is performed 20 times using the same input contact data, but starting from different random initial structures to finally generate a structural ensemble. Si-C combines the advantages of ISD and NucDynamics and enables rapid inference of biologically and statistically valid whole-genome structure ensemble at unparalleled high resolution from single-cell Hi-C data of mammalian cell.

## Results

In this study, we applied the Si-C approach to structure determinations from the single-cell Hi-C data of eight G1-phase haploid mouse embryonic stem (ES) cells[9] (data are downloaded from Gene expression Omnibus or GEO with accession code GSE80280). For comparison, structure reconstructions were also performed by the NucDynamics and SCL methods based on the same data. Before using the Hi-C data to model 3D structures either by Si-C, NucDynamics, or SCL, we removed isolated

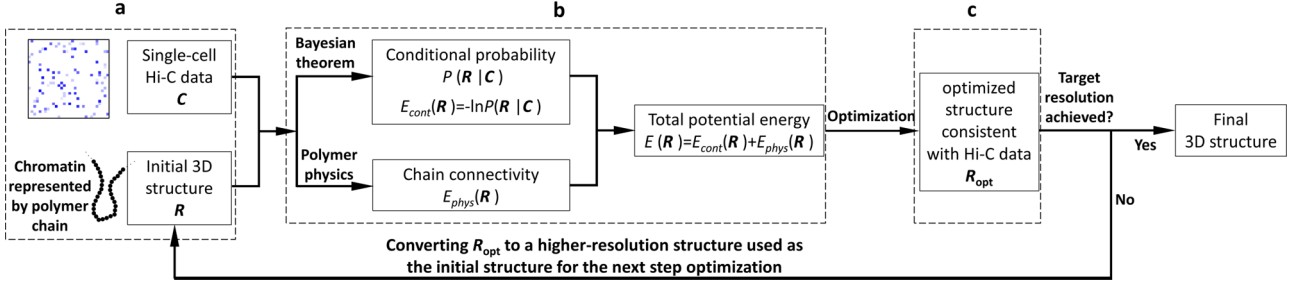

**Fig. 1 Schematic illustration of Si-C method for inferential structure determination from single-cell Hi-C data. a** Building models of chromosomes by using polymer chains consisting of contiguous, equally sized beads, and assigning the experimentally observed contacts to the pair of chromatin beads containing the corresponding restriction fragment ends. **b** Defining the total potential energy of a 3D conformation. **c** Performing a structure optimization by using the steepest gradient descent algorithm from an initial conformation to minimize $E(\mathbf{R})$.

contacts that are not supported by other contacts between the same two regions of 2 Mb because these contact reads have a high risk of sequence mapping errors. Details are presented in "Methods" section and Supplemental Notes 1 and 2, respectively.

**The percentage of contact restraints that are violated in the 3D model**. Since both Si-C and NucDynamics are data-driven methods, the consistence of their models with the experimental contact constraints should be a requisite to assess the validations of the two methods. As a first validation of Si-C, we measured the percentage of experimental contact restraints, which are violated in the calculated 3D structures. Contact restraints are violated if the corresponding two beads are separated with a distance >2 bead diameters (bds). Figure 2a shows that the average violation percentages of the Si-C 100-kb resolution structure ensembles of the eight cells are all <1%, while those of the published NucDynamics 100-kb structure ensembles of the same eight cells (structures are downloaded from GEO with accession code GSE80280) range from ~5 to ~11%, except for cell 8. Moreover, the Si-C structures at 10 and 1 kb resolution also show very low violation percentages. We further ran NucDynamics to generate the structure ensemble of 10 kb resolution for cell 1 and, however, find its average violation percentage to be >70%. In terms of the consistence between models and the contact constraints, the Si-C models of 100, 10, and even 1 kb are reliable.

**Validation with 3D-FISH**. Beagrie et al. reported the distances detected by eight fluorescence in situ hybridization (FISH) probe pairs that are located on chromosomes 3 and 11[22]. As a second validation, we calculated the correlation between the distances reported by Beagrie et al. and the corresponding distances in the Si-C structures (details are presented in Supplementary Note 4). As shown in Fig. 2b, the Si-C structure of 10 kb resolution shows a high correlation with the FISH data, with Pearson correlation coefficients of 0.889. At 100 kb resolution, the correlation of the Si-C structure is 0.931 and that of the published NucDynamics structure is 0.888 (see Supplementary Fig.1). The high correlation strongly validates the Si-C method.

**Conformation variability within the calculated structure ensemble**. We assessed the degree of conformation variability within the same structure ensemble by pairwise root-mean-square deviation (RMSD)[23]. The details of RMSD calculation are described in Supplementary Note 5. For cells 1, 2, and 3, Fig. 2d shows that the variability in the Si-C ensembles at three resolutions (median RMSDs < 0.1 nuclear radii) are comparable to the published NucDynamics 100-kb structure ensembles[9]. However, the Si-C method seems to produce very heterogeneous ensembles

for cells 6 and 7 in terms of the values of RMSDs. Especially, the median RMSD of the Si-C 100-kb structure ensemble of cell 6 is >0.2 nuclear radii.

For the case of 3D structure reconstruction of a chromosome that is represented by polymer chain consisting of numerous beads, high RMSD between two conformations might be caused by the flexibility of the polymer chain, the inconsistence in chromatin folding between two conformations, or both of them. To investigate the rationales behind high RMSDs within the Si-C ensemble, we selected two conformations of chromosome 1 from the 100-kb structure ensemble of cell 6 (the RMSD between them is as high as 0.29 nuclear radii) and displayed their 3D structures in Fig. 2e. A quick glance shows that both of the two structures can be viewed to consist of two parts that are connected by a flexible chain and the relative locations of the two parts in the two conformations are apparently different. To quantitatively assess the difference in the chromatin folding between the two conformations, we computed a distance matrix for each conformation (see Fig. 2f; details of distance matrix calculation are presented in Supplementary Note 6). The two matrices show a high correlation with each other, with Pearson correlation coefficient of 0.96 (Fig. 2g). The high correlation indicates that the flexibility of the polymer chain predominantly contributes to the high RMSD between the two conformations. In other words, the results demonstrate that the Si-C method enables to produce highly consistent structures in terms of chromatin folding.

Furthermore, it should be emphasized that the RMSDs of NucDynamics displayed in Fig. 1d are calculated from the published structures[9]. We also directly ran the NucDynamics code downloaded from ref. [9] using the same input Hi-data as these for Si-C to generate structure ensembles for all eight cells and referred to these structures as the calculated NucDynamics structures in order to distinguish them from the published NucDynamics structures. The RMSDs of the calculated NucDynamics ensemble are substantially comparable to the Si-C RMSD for each cell (see Supplementary Fig. 2). The inconsistence between the published and the calculated NucDynamics structures originates from the lack of the best-model selection process and full ambiguous protocol in the NucDynamics code of the version in ref. [9], which are performed for generating the published NucDynamics structures.

As shown in Fig. 2d, the Si-C ensembles at different resolutions of the same cell exhibit similar structural variability. For instance, the median RMSDs of the Si-C 100-, 10-, and 1-kb resolution structure ensembles are 0.02, 0.04, and 0.06 nuclear radii for cell 1 and 0.21, 0.12, and 0.18 nuclear radii for cell 6, respectively. The small differences between median RMSDs of the Si-C ensembles at different resolutions of the same cell support the validity of the Si-C approach.

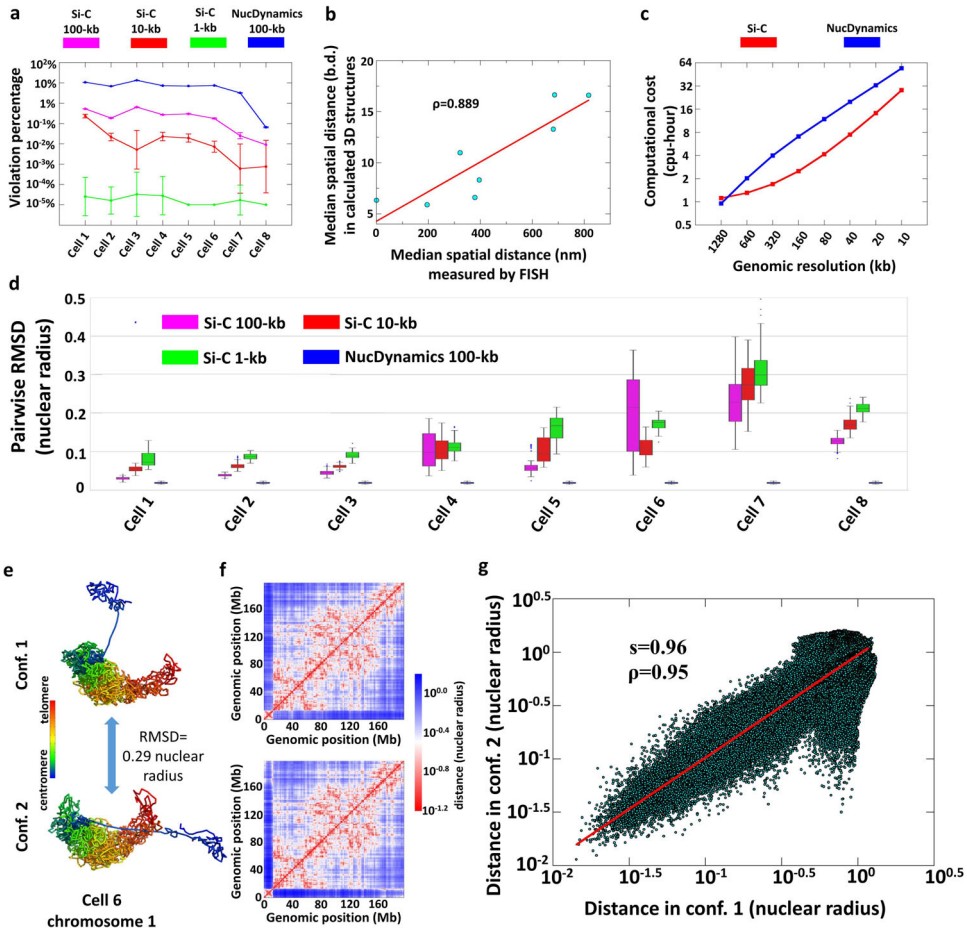

**Fig. 2 Estimating the validity and performance of Si-C. a** The average percentages of experimental contact restraints violated in the Si-C ensembles at 100 kb (purple), 10 kb (red), and 1 kb (green) resolutions along with the average violation percentages of the published NucDynamics 100-kb structure ensembles of the same cells (blue) (structures downloaded from GEO with accession code GSE80280). The distance threshold to decide whether two beads are in contact set to 2 bead diameters. Data are presented as mean ± SD. For each structure ensemble generated by Si-C, 20 structure replicas are used in the statistical analysis. For structure ensemble downloaded from GEO, ten structure replicas are used. **b** Correlation between the median spatial distances measured by eight 3D-FISH probe pairs (from ref. [22]) and the median distances of the corresponding pairs in the Si-C inferred structures of 10 kb resolution. Details for calculating the median spatial distances are shown in Supplement Note 4. **c** Comparison of Si-C (red) and NucDynamics (blue) in terms of computational cost for structure reconstruction for cell 1. Details are presented in Supplementary Note 3. **d** Boxplot of pairwise RMSDs within the Si-C ensembles at 100 kb (purple), 10 kb (red), and 1 kb (green) resolutions along with pairwise RMSDs within the published NucDynamics ensembles of the same cells (blue) (structures downloaded from GEO with accession code GSE80280). Median values are shown by black bars. Boxes represent the range from the 25th to the 75th percentile. The whiskers represent 1.5 times the inner quartile range. For each structure ensemble generated by Si-C, 20 structure replicas are used in the statistical analysis. For structure ensemble downloaded from GEO, ten structure replicas are used. **e** The 10 kb structures of two conformations of chromosome 1 of cell 6 with the RMSD between them of 0.29 nuclear radii. Chromosome regions are colored from blue to red (centromere to telomere). **f** The distance matrices derived from the two 3D structures shown in **e**. **g** Correlation between the two distance matrices in (**f**). The Pearson correlation coefficient ($\rho$) is 0.95. The red line is power-law fit with a scaling exponent (s) equal to 0.96. RMSD root-mean-square deviation.

**Agreement between the experimental and back-calculated data.** To quantitatively evaluate the agreement between 3D models and Hi-C maps, we translated 3D models into distance matrices (details are presented in Supplementary Note 6). For example, we constructed the distance matrices for the 10-Mb genomic regions (Chr1: 30–40 Mb) based on the Si-C 10-kb structures of the eight individual cells (see Fig. 3a) and averaged them across the eight cells to obtain an average distance matrix (see Fig. 3c) that can be compared to the population Hi-C contact-frequency matrix[24] (Fig. 3b; populated Hi-C data downloaded from GEO with accession code GSE35156). The spatial distance in Fig. 3c displays a high correlation with the Hi-C contact frequency in Fig. 3b, with Pearson correlation coefficient of −0.92 (see Fig. 3d). Furthermore, the average distance matrix (see Fig. 3c) shows domain structures similar to topological-associated domains (TADs)

observed in the population Hi-C matrix (see Fig. 3b). We refer to these domain structures as TAD-like structures and identify their boundaries using separation score (details of separation score calculation are presented in Supplementary Note 7). As shown in Fig. 3f, the positions corresponding to the separation score peaks are considered to be boundaries of TAD-like domains. Figure 3e shows the positions of TAD boundaries in the population Hi-C map, which are identified by the TopDom algorithm[25] (details for TAD boundary identification are presented in Supplementary Note 7). Comparison between Figs. 3f, e shows that most of the boundaries of TAD-like structures align with the positions of TAD boundaries. These findings demonstrate the significant similarity between the calculated average distance matrix and the population Hi-C contact-frequency matrix, strongly validating the 10-kb resolution structures generated by Si-C.

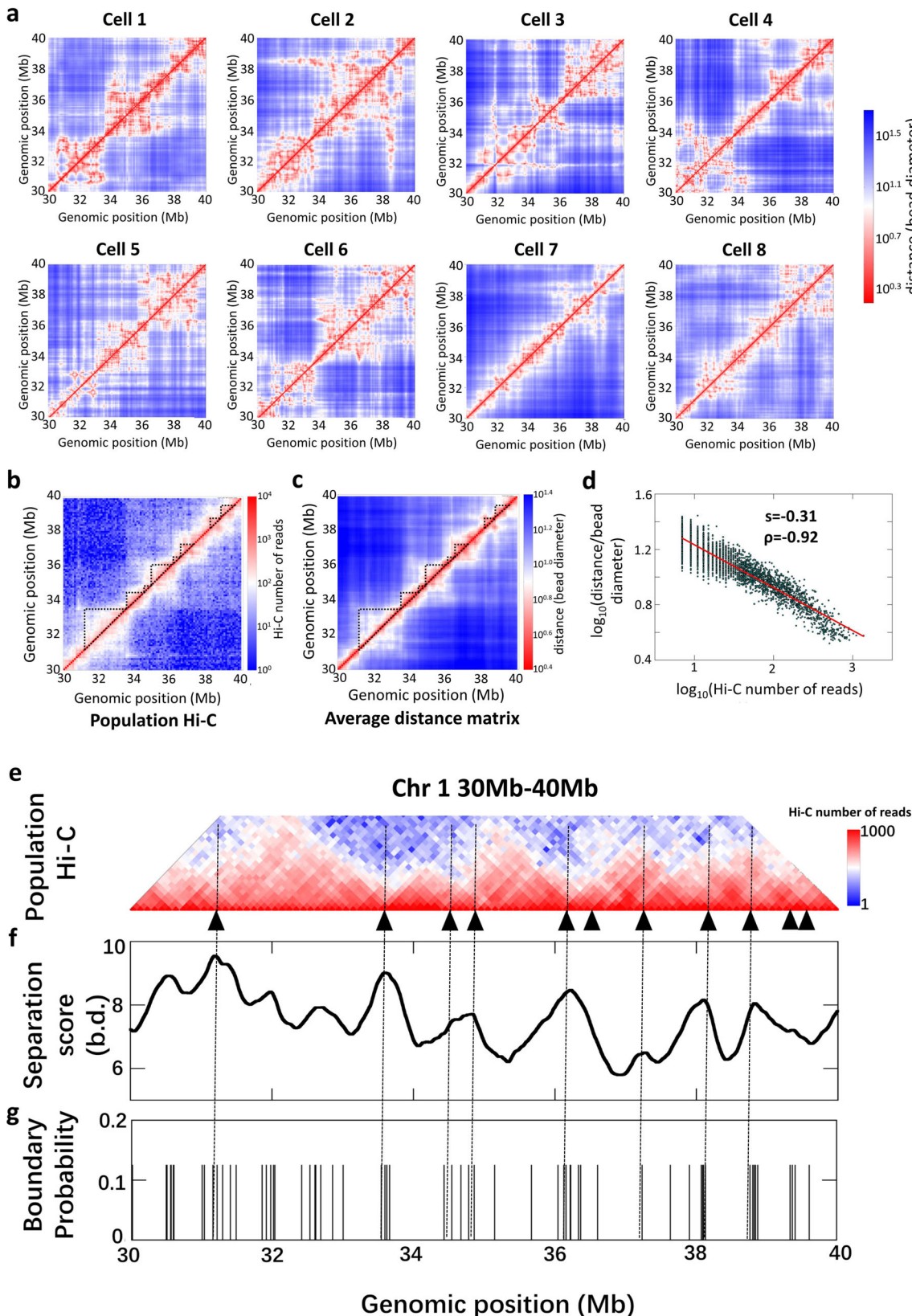

A closer inspection of Fig. 3a immediately shows that the domain structure is also a fundamental hallmark of the genome organization of individual cells, and the sizes and locations of these domains display substantial cell-to-cell variations. These findings are consistent with previous super-resolution imaging experiment[7]. To carry out a quantitative comparison with the observation of the super-resolution imaging experiment[7], we also identified domain boundaries in the eight matrices shown in Fig. 3a and measured the probability for each genomic position appearing as a single-cell domain boundary throughout the eight matrices. Notably, the 10-Mb region is represented by 1000 beads at the 10 kb resolution, among which only ∽6% of beads appear

**Fig. 3 Si-C calculated structures allow identification of TADs in the population Hi-C contact-frequency matrix. a** The 10-kb resolution distance matrices for the 10-Mb genomic regions (Chr1: 30–40 Mb) of the eight individual mouse ES cells calculated from the corresponding Si-C 10-kb resolution structures. **b** The 100-kb resolution population Hi-C contact-frequency matrix of mouse ES cells for the same 10-Mb genomic region (Chr1: 30–40 Mb) (data are downloaded from GEO with accession code GSE35156). TADs identified previously are marked by triangles. **c** The 100-kb resolution average distance matrix is derived from the eight 10-kb resolution distance matrices shown in (**a**). TAD-like regions identified by separation score are marked by triangles. **d** Correlation between the spatial distance and Hi-C contact-frequency shown in (**c**, **b**). The Pearson correlation coefficient ($\rho$) is −0.92. The red line is power-law fit with a scaling exponent equal to −0.31. **e** Enlarged view of (**b**) with the positions of TAD boundaries labeled by black triangles. **f** Separation score for each genomic position in (**c**). bd bead diameter. **g** Probability for each genomic position to appear as a single-cell domain boundary. Single-cell domain boundaries are identified by the computing separation score for each genomic position of each distance matrices shown in (**a**).

as a single-cell boundary in any of eight cells with nearly the same probability of 0.125 (Fig. 3g). Whilst the number of cells is small, the statistical results presented in Fig. 3g demonstrate two pronounced trends. First, the number of positions that appear as a single-cell domain boundary is evidently higher than the total number of established TAD boundaries in the populated Hi-C matrix (Fig. 3e). Second, the regions where the single-cell boundaries are enriched are well aligned with the positions of TAD boundaries. These trends are in agreement with the observations of the super-resolution chromatin tracing experiment[8] that single-cell domain boundaries occur with nonzero probability at all genomic positions, but preferentially at positions that are identified as TAD boundaries in the Hi-C contact-frequency map[8]. The agreement provides another support for the validities of the 10-kb resolution Si-C structures.

Furthermore, we mapped the 3331 CTCF/cohesion loops that are detected from the high-resolution Hi-C data of mouse B lymphoblasts[26] onto the eight Si-C 10-kb structures, and found that <3% of the loops are formed in each individual cells and the percent obviously varies from cell to cell (see Supplementary Fig. 3). The results demonstrate that CTCF/cohesion loops detected from the bulk data do not form in all individual cells, which is consistent with previous observations from DNA-FISH[26] and analysis based on NucDynamics method[9].

**Conserved 3D whole-genome architecture in all cells.** Several conserved features of 3D genome architecture are consistently observed in the Si-C structures at different resolutions for all eight G1-phase haploid ES cells. Figure 4a, b displays the features of cell 1, while those of the other seven cells are shown in Supplementary Fig. 4. First, Fig. 4a shows that individual chromosomes occupy distinct territories with a degree of chromosome intermingling (Supplementary Fig. 5, calculation details are presented in Supplementary Note 8), and meanwhile centromeres and telomeres localize at the opposite sides of the nucleus (Supplementary Fig. 6). Second, each territory is divided into euchromatic and heterochromatic regions, known as A and B compartments (Fig. 4c), which is consistent with imaging experiments[27]. (Details of compartment identification are presented in Supplementary Note 9). Third, all chromosomes stack together to give a sphere consisting of two shells and a core: an outer B compartment shell, an inner A compartment shell, and an internal B compartment hollow core (at the 100 kb resolution) or seemingly solid core (at the 10 or 1 kb resolution) of B compartment (see Fig. 4b). Mapping various experimental data, such as lamina-associated domain[28] (LAD), CpG density, and DNA replication timing[29] (data are obtained from ArrayExpress with accession code E-MTAB-3506), onto the 3D models shows similar architectures (Fig. 4b), demonstrating that active segments are predominately located in the inner A compartment shell, while inactive segments mostly distribute in the outer or internal B compartment regions. These conserved features of intact genome architecture showing in the Si-C structures are consistent with various independent observations[30–33] and analysis based on

the NucDynamics models, demonstrating that Si-C enables to infer biologically reliable 3D genome structures from single-cell Hi-C data.

Figure 4d displays another example to illustrate an independent evaluation of the validity of Si-C. Although the features of 3D genome architectures shown in Fig. 4a–c are conserved in all eight cells, the individual chromosome structure varies remarkably from cell to cell (Fig. 4d). The cell-to-cell variation is in agreement with the observation of previous single-cell experiments[8,9,16].

**Unparalleled details of chromatin folding provided by Si-C models.** We showed details of chromatin folding based on the Si-C 10-kb 3D structures and compared them with the Si-C100-kb structures and the published NucDynamics 100-kb structures. Figure 5 displays the 10-kb resolution 3D structures of four different regions, each of which includes two neighboring domains. It can be seen that chromatin forms distinct spatially segregated globular domain structures (Fig. 5b), which is supported by the super-resolution imaging experiment[8]. Notably, domains are linked by highly extended chromatin chains. In other words, the 3D structures of boundary regions correspond to extended chromatin chains (Fig. 5a). On examining the 3D structures in Fig. 5b, it is difficult to judge whether chromatin fibers of domain regions are uniformly compacted or not. To quantitatively estimate the degrees of compaction, we computed the gyration radius ($R_g$), which is defined as the root-mean-square distance of bead positions in each 200 kb fragment from the centroid of the 200-kb fragment (details of calculating $R_g$ are presented in Supplementary Note 10), and we assigned the value of $R_g$ to the midpoint bead of the 200-kb fragment. After taking every bead in the four 10-Mb regions as a midpoint of a 200-kb fragment, we calculated the $R_g$ for a total number of 4000 fragments and obtained the numerical distributions of $R_g$ for the four regions (see the bottom of Fig. 5a). The peaks in the distributions of $R_g$ correspond to the regions that show a higher degree of extension than their neighborhood. It can be seen that each boundary region indeed aligns to a peak. Significantly, there are many peaks locating in domain regions, implying that the domain is not a uniformly compacted structure and there might be detailed structures.

In Fig. 5c, d, we show the loop and hairpin-like hyperfine structures as examples of detail structures embodied in domain regions. For comparison, 100-kb resolution models of the same regions are also displayed. It can be seen that loops or hairpin-like structures are obscured or even cannot be visualized in the 100-kb resolution structures inferred either by Si-C or NucDynamic. Obviously, the analysis of the 10-kb resolution structures can provide unparalleled details of chromatin folding.

**High computational efficiency of Si-C.** Last, we compared the execution speeds of Si-C and NucDynamics for reconstructing intact genome structures from the same single-cell Hi-C data at different resolutions (see Fig. 2c; details of computing resources

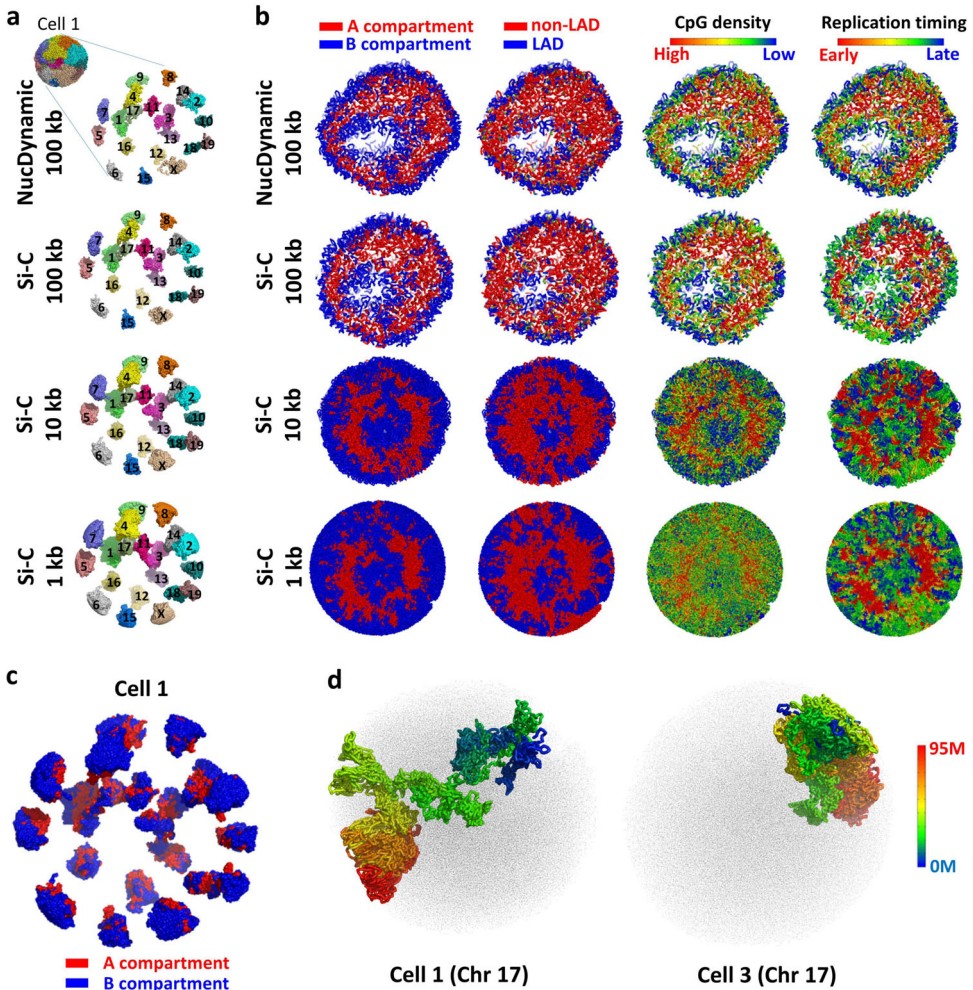

**Fig. 4 Large-scale 3D structure of the genome. a** The Si-C calculated intact genome 3D structures of cell 1 at 100, 10, and 1 kb resolution with an expanded view of the separated chromosome territories along with the published NucDynamics 100-kb resolution structure of cell 1. The 100-kb resolution structures of NucDynamics are downloaded from GEO with accession code GSE80280. **b** Cross-sections of intact genome 3D structures of cell 1, colored according to whether the sequence is in the A (red) or B (blue) compartment (first column); whether the sequence is part of a lamina-associated domain (LAD) (blue) or not (red) (second column); the CpG density from red to blue (high to low) (third column); the replication time in the DNA duplication process from red to blue (early to late) (fourth column). **c** The Si-C calculated 10-kb resolution intact genome 3D structure of cell 1 with an expanded view of the spatial distribution of the A (red) and B (blue) compartments. **d** The Si-C calculated 10-kb resolution structures of chromosome 17 of cells 1 and 3 colored from red to blue (centromere to telomere) with structures of other chromosomes shown as black dots.

are presented in Supplementary Note 3). The comparison shows that, except for 1280 kb resolution, the execution speed of Si-C is twice as fast as that of NucDynamics. Specifically, it costs 28 cpu-h for Si-C to infer a 10-kb resolution intact genome structure for cell 1, while it is 54.3 cpu-h for NucDynamics.

We also compared the execution speeds of Si-C with SCL. The time consumed by SCL to reconstruct the structure of chromosome 1 of cell 1 at the 100 kb resolution is nearly equal to that for reconstructing intact genome structure of 100 kb resolution by Si-C.

## Discussion

In this work, we proposed a Bayesian probability-based method, Si-C, to infer the 3D genome structure (**R**) with the highest probability in the light of sparse and noisy single-cell Hi-C data (**C**). Compared with the NucDynamics approach, which is the method that has the ability to reconstruct whole-genome structures from single-cell Hi-C data of mammalian cells, Si-C substantially outperformed NucDynamics at rapidly determining biologically and statistically valid super-resolution whole-genome

3D structure. The superiority of Si-C can be attributed to the following. First, Si-C directly describes the single-cell Hi-C contact restraints using an inverted-S-shaped probability function of the distance between the contacted locus pair, instead of translating the binary contact into an estimated distance. For locus pairs without contact reads, an S-shaped probability function is used to describe the cases where two loci are spatially close to each other, but the proximity is not detected in the Hi-C experiment. The statistical foundation makes Si-C efficient for structure determination from extremely sparse and noisy data. Second, Si-C adopts the steepest gradient descent algorithm to maximize the conditional probability $P(\mathbf{R}|\mathbf{C})$, which allows us to a rapid inference of an intact genome structure from single-cell Hi-C data of mammalian cells at resolution varying from 100 to 1 kb.

For a given single-cell Hi-C dataset, when the resolution of the calculated structure improves, the number of beads that are not constrained by experimental data increases, and then the overall precision of inferred structure decrease. Naturally, a question is raised: at least how many contact reads are required for the Si-C

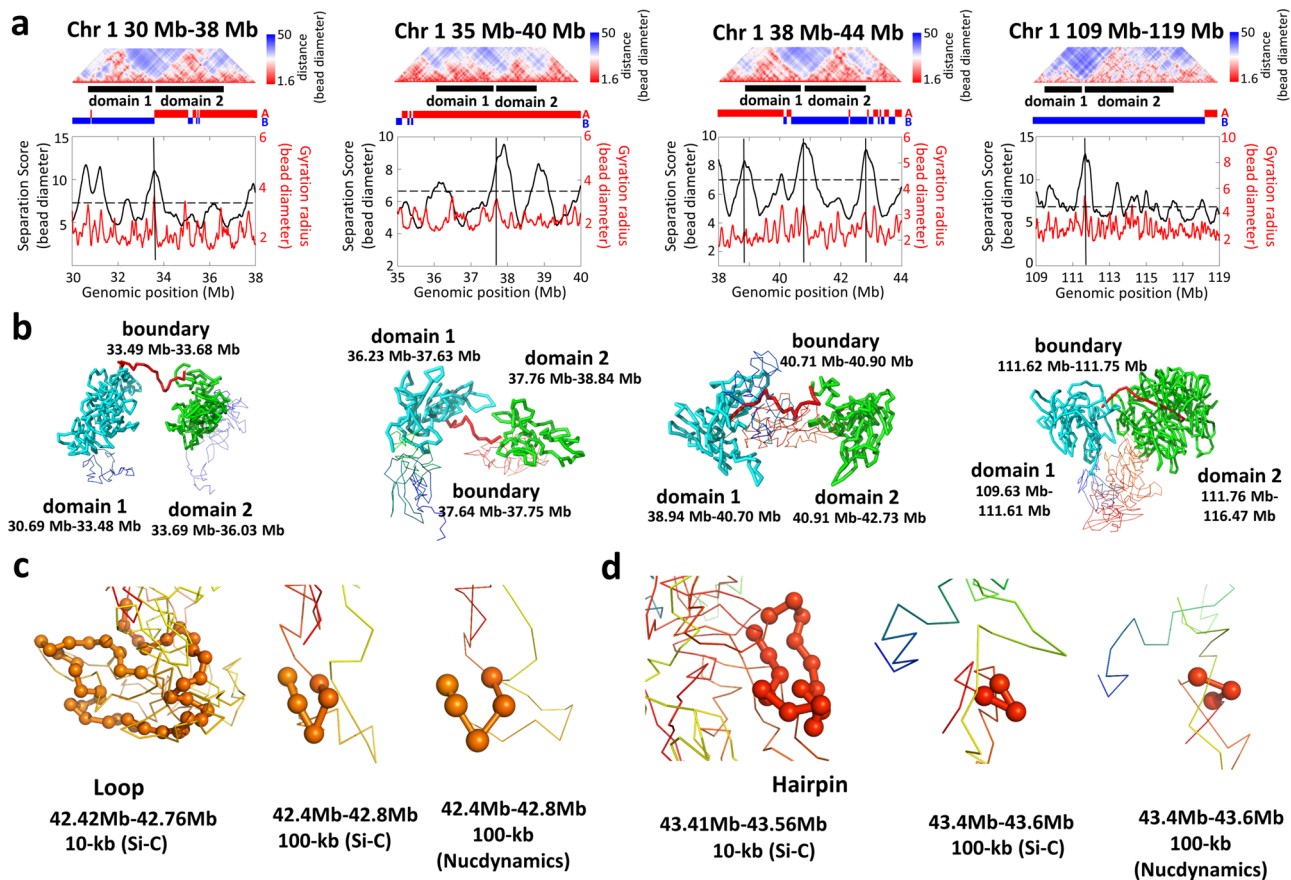

**Fig. 5 Small-scale 3D structure at 10 kb resolution. a** Top: heatmap plot of distance matrices derived from the Si-C 10 kb structure of cell 1 for four genomic regions. In each region, two neighboring domains that are identified by separation score are indicated by black lines. Bottom: distributions of the separation score and gyration radius ($R_g$). **b** 3D structures of the genome regions associated with the two neighboring domains shown in (**a**). Structures corresponding to the boundary, domain 1 and domain 2 regions are shown in red, cyan, and green, respectively. **c, d** Loop and hairpin-like hyperfine structures visualized by beads of 10 kb size (left) and the 3D structures of the same regions visualized by beads of 100 kb size, inferred by Si-C (middle) and Nucdynamics(right).

method to achieve a reliable 3D structure. Since the median of pairwise RMSDs within the calculated structure ensemble fluctuates along with the change in resolution, we addressed the question by analyzing the relationship between resolution and the median of pairwise RMSDs within the Si-C inferred structure ensemble. We assumed that an inferred ensemble is reliable if its median value of pairwise RMSDs is <4 bds. The motivation behind the suggestion of 4 bds originates from the distance threshold between a contacted bead pair. The distance threshold is set to 2 bds, and thus one bead of the contacted bead pair can move in the scope of a sphere of diameter 4 bds, relative to the other bead. Figure 6 shows the plot of the median value of pairwise RMSDs within structure ensembles of cell 1 against the resolution. According to the threshold of 4 bds, Fig. 6 demonstrates that the highest attainable resolution is 4 kb. At such resolution, the whole genome of cell 1 is divided into 658,453 beads, while the number of Hi-C contact read is 110,623 (each bead corresponds to an average of 0.168 contact reads). Therefore, to obtain a reliable 3D genome structure, we suggest that the resolution should be selected such that the amount of contact reads per chromatin bead should be > 0.2.

In conclusion, our work shows that Si-C is a biologically and statistically viable method, allowing us to rapidly infer a reliable 3D structure at super-resolution, which provides unprecedented details on chromatin folding.

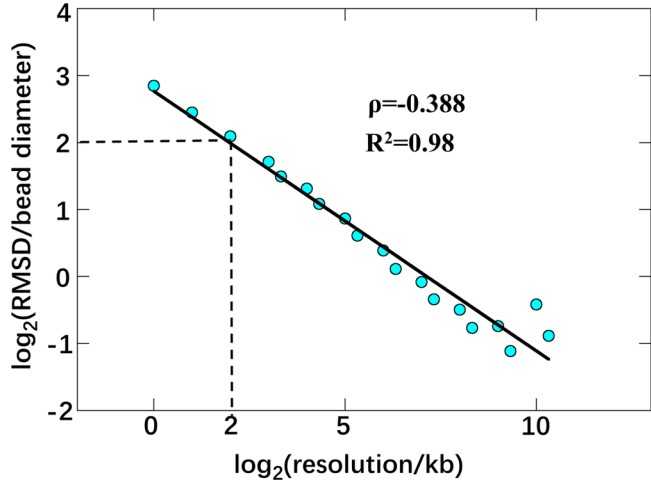

**Fig. 6 Estimating how many contact reads per chromatin bead are required for Si-C to achieve reliable 3D structure.** Log-scale plot of median RMSD within the structure ensemble of cell 1 against the resolution. The threshold of the median value of pairwise RMSDs for a reliably calculated structure ensemble set to be 4 bead diameters, indicated by the dashed line. RMSD root-mean-square deviation.

## Methods

**Bead-on-a-string representation of chromosome.** Each chromosome is represented as a polymer chain consisting of contiguous, equally sized beads with a diameter of $a$. The connectivity between adjacent beads $(i, i+1)$, which are in the same chromosome, is described by the harmonic oscillator potential:

$$\varepsilon_{\text{phys}}\left(r_{i,i+1}\right) = \frac{k}{2}\left(r_{i,i+1} - a\right)^2 \tag{1}$$

where the force constant $k$ is set to $100/a^2$ and $r_{i,i+1}$ is the Euclidean distance between the $i$th and $i+1$th beads. It should be noted that diameter $a$ is used as the unit length in our method.

**Derivation of conditional probability.** The basic aim of Si-C is to find out the 3D genome structure ($R$) with the highest probability in the light of the contact restraints ($C$) derived from single-cell Hi-C contact data. Formally, such structure determination can be solved by the probability distribution $P(\mathbf{R}|\mathbf{C})$ defined over the complete conformation space. In our approach, the conditional probability $P(\mathbf{R}|\mathbf{C})$, also called posterior probability, is defined as:

$$P(\mathbf{R}|\mathbf{C}) \propto \prod_{i<j} P\left(r_{ij}|c_{ij}\right) \tag{2}$$

where $r_{ij}$ is the Euclidean distance between pair of the $i$th and $j$th beads, denoted as beads $(i,j)$, and $c_{ij}$ is the count of detected contact reads between beads $(i,j)$ in the single-cell Hi-C matrix. According to the Bayesian theorem, the posterior probability $P\left(r_{ij}|c_{ij}\right)$ can be factorized into two components:

$$P\left(r_{ij},|,c_{ij}\right) \propto P\left(c_{ij},|,r_{ij}\right) P\left(r_{ij}\right) \tag{3}$$

where $P\left(c_{ij},|,r_{ij}\right)$ is the likelihood and $P\left(r_{ij}\right)$ the prior probability.

The likelihood $P\left(c_{ij},|,r_{ij}\right)$ quantifies the consistence between the contact data and the structural model. It is the probability of observing $c_{ij}$ contacts between beads $(i,j)$ when they are spatially separated by the distance of $r_{ij}$. Based on the value of $c_{ij}$, all bead pairs can be divided into two groups. In one group where the value of $c_{ij}$ is not equal to zero ($c_{ij} \neq 0$), the beads $(i,j)$ is restrained in close spatial proximity. The $P\left(c_{ij}\neq 0,|,r_{ij}\right)$ would decrease with the increase in $r_{ij}$. Therefore, when $r_{ij}$ is over than the distance threshold $r_0$, the corresponding $P\left(c_{ij}\neq 0,|,r_{ij}>r_0\right)$ would be zero. Here, we use an inverted-S-shaped probability function to transform the contact constraints derived from single-cell Hi-C data into a function of the distance. The inverted-S-shaped probability function is

$$P\left(c_{ij}\neq 0,|,r_{ij}\right) \propto \begin{cases} \left(\frac{r_0 - r_{ij}}{r_0 + r_{ij}}\right)^{\frac{c_{ij}}{2}} & \text{for } c_{ij}\neq 0 \text{ and } r_{ij}<r_0 \\ 0 & \text{for } c_{ij}\neq 0 \text{ and } r_{ij} \geq r_0 \end{cases} \tag{4}$$

where the distance threshold $r_0$ is set to $2a$.

In the other group where the value of $c_{ij}$ is equal to zero ($c_{ij} = 0$), the absence of contact read between the beads $(i,j)$ cannot be interpreted only as far separation between the beads $(i,j)$. There are two distinct rationales behind the absence. The first one is that the absence is indeed because the beads $(i,j)$ are too far apart, namely $r_{ij} \geq r_0$. Therefore, $P\left(c_{ij} = 0,|,r_{ij}>r_0\right)$ approximately equals to 1. The second case is that the beads $(i,j)$ are closely located but not detected. This situation is more likely to occur for the pairs in which beads are separated by a large distance, but still within the threshold $r_0$. For this situation, the likelihood $P\left(c_{ij} = 0,|,r_{ij}<r_0\right)$ is modeled by an S-shaped probability function:

$$P\left(c_{ij} = 0,|,r_{ij}<r_0\right) \propto \left(\frac{1}{1+e^{-4(r_{ij}-r_0)}}\right)^{0.075} \text{ for } c_{ij} = 0 \tag{5}$$

The power coefficient 0.075 is an arbitrary value used in this formula to increase the probability.

The prior probability $P\left(r_{ij}\right)$ describes the probability of the distance $r_{ij}$ between beads $(i,j)$ based on our prior knowledge. Assuming chromatin beads are uniformly distributed in nuclear space, the prior probability $P\left(r_{ij}\right)$ is proportional to the surface area of the sphere with radius of $r_{ij}$. Therefore, the $P\left(r_{ij}\right)$ is defined as:

$$P\left(r_{ij}\right) \propto r_{ij}^2 \tag{6}$$

Finally, the posterior probability $P\left(r_{ij},|,c_{ij}\right)$ is adopted as follows:

$$P\left(r_{ij},|,c_{ij}\right) \propto \begin{cases} r_{ij}^2\left(\frac{1}{1+e^{-4(r_{ij}-r_0)}}\right)^{0.075} & \text{for } c_{ij} = 0 \\ r_{ij}^2\left(\frac{r_0 - r_{ij}}{r_0 + r_{ij}}\right)^{\frac{c_{ij}}{2}} & \text{for } c_{ij} \neq 0 \text{ and } r_{ij} < r_0 \\ 0 & \text{for } c_{ij} \neq 0 \text{ and } r_{ij} \geq r_0 \end{cases} \tag{7}$$

**Total potential energy of a 3D conformation.** We define the total potential energy of a 3D conformation with two terms, one of which describes the contact restraints and the second is the above-mentioned harmonic oscillator potential introduced to ensure the connectivity of the chromosome backbone. The negative logarithm of the posterior probability, namely $-\ln P\left(r_{ij},|,c_{ij}\right)$, is analogous to a physical energy, denoted as $\varepsilon_{\text{cont}}\left(r_{ij}, c_{ij}\right)$. The $\varepsilon_{\text{cont}}\left(r_{ij}, c_{ij}\right)$ is used to describe the potential energy originated from the contact restraints. Combining $\varepsilon_{\text{cont}}\left(r_{ij}, c_{ij}\right)$ with the harmonic oscillator potential $\varepsilon_{\text{phys}}\left(r_{i,i+1}\right)$, the total potential energy $E(\mathbf{R})$ of a 3D conformation is calculated according to the formula below:

$$E(\mathbf{R}) = \sum_{i,i+1 \text{ in the same chr.}} \varepsilon_{\text{phys}}\left(r_{i,i+1}\right) + \sum_{i<j}\varepsilon_{\text{cont}}\left(r_{ij}, c_{ij}\right) \tag{8}$$

$$\varepsilon_{\text{cont}}(r_{ij}, c_{ij}) = \begin{cases} -2\ln r_{ij} + 0.075\ln(1 + e^{-4(r_{ij}-r_0)}) \text{ for } c_{ij} = 0 \\ -2\ln r_{ij} + \frac{c_{ij}}{2}\ln(\frac{r_0 - r_{ij}}{r_0 + r_{ij}}) \text{ for } c_{ij}\neq 0 \end{cases} \tag{9}$$

It should be noted that, according to Eq. (7), when $r_{ij}$ increases towards or beyond the distance threshold $r_0$ for the case of $c_{ij}\neq 0$, the posterior probability decreases to 0 that makes $-\ln P\left(r_{ij},|,c_{ij}\right)$ nonsense. To address this issue, we employ Taylor expansion to formulate the energy $\varepsilon_{\text{cont}}\left(r_{ij}>0.995r_0, c_{ij}\neq 0\right)$ as the following:

$$\varepsilon_{\text{cont}}\left(r_{ij}>0.995r_0, c_{ij}\neq 0\right) = \varepsilon_{\text{cont}}\left(r_{ij} = 0.995r_0, c_{ij}\neq 0\right)$$
$$+ \frac{\partial\varepsilon_{\text{cont}}\left(r_{ij} = 0.995r_0, c_{ij}\neq 0\right)}{\partial r_{ij}}\left(r_{ij} - 0.995r_0\right) \tag{10}$$

For the bead-on-a-string model of a whole-genome structure, the number of bead pairs belonging to the group of $c_{ij} = 0$ is huge. To reduce the computational cost, we set the $\varepsilon_{\text{cont}}\left(r_{ij}>3a, c_{ij} = 0\right)$ equal to the $\varepsilon_{ij}\left(r_{ij} = 3a, c_{ij} = 0\right)$ during the calculation.

**Maximization of conditional probability.** 3D genome structure optimization is performed by using the steepest gradient descent algorithm from an initially random conformation to maximize the conditional probability $P(\mathbf{R}|\mathbf{C})$ and output the corresponding structure that is compatible with the experimental contact restraints. In each optimization step, the force that acts on the $i$th bead, denoted as $\vec{f}_i$, is firstly calculated as the negative derivative of potential energy on the bead coordinate,

$$\vec{f}_i = -\frac{\partial E(\mathbf{R})}{\partial \vec{r}_i} \tag{11}$$

Then, the displacement $\vec{d}_i$ is set to be proportional to the calculated force,

$$\vec{d}_i = \frac{0.1a}{f_{\max}}\vec{f}_i \tag{12}$$

in which $a$ is the bead diameter and $f_{\max}$ is the maximum force among the forces on every bead. In the first 2000 optimization steps, the modeled structure is shrunk once per ten steps. The structure shrinking is performed by multiplying the coordinates of all beads with a factor $\left(\frac{\sum_{i,j,c_{ij}\neq 0} a}{\sum_{i,j,c_{ij}\neq 0} r_{ij}}\right)^{0.05}$, in which $r_{ij}$ is the distance between the $i$th and $j$th beads and $a$ is the bead diameter. After the first 2000 optimization steps, another 8000 optimization steps without structure shrinking are carried out to output the finally optimized structure.

**Hierarchical optimization strategy.** The optimized structure of low resolution is used to generate the initial structure of high resolution for another optimization procedure to obtain high-resolution optimized structure. There are two steps for deriving a high-resolution initial structure from the optimized structure of low resolution. First, each bead of the optimized structure of low resolution is evenly divided into two small beads. Second, based on the coordinates of the $i$th and $i+1$th parent beads in the low-resolution structure, denoted as $\vec{r}_{i,\text{low}}$ and $\vec{r}_{i+1,\text{low}}$, we define the coordinates of the two small derived beads from the $i$th parent bead,

denoted as $\vec{r}\,'_{i,\text{high}}$ and $\vec{r}\,''_{i,\text{high}}$, according to the following:

$$\vec{r}\,'_{i,\text{high}} = \vec{r}_{i,\text{low}} \tag{13}$$

$$\vec{r}\,''_{i,\text{high}} = \frac{\left(\vec{r}_{i,\text{low}} + \vec{r}_{i+1,\text{low}}\right)}{2} \tag{14}$$

From the 3D coordinates of all the small derived beads, further 10,000 optimization steps without structure shrinking are performed to minimize the potential energy $E(\mathbf{R},\mathbf{C})$ and output the optimized structure of high resolution. We continue the cycle until achieving the optimized structure at a desired resolution. For each single cell, the whole calculation is repeated 20 times based on the same experimental restraints, but starting from different random structures, thereby producing 20 structure replicas.

Here, we present the steps for the calculation of a 10-kb resolution structure as an example. First, the initial whole-genome structure consisting of beads representing 1280-kb regions of chromosome sequence was randomly generated and then optimized following the procedure described in the section "Maximization of conditional probability." After that, the optimized 1280-kb resolution structure was transformed into a 640-kb resolution structure by dividing each 1280 kb bead into two 640 kb beads. Such 640-kb resolution structure was used as the initial structure for the calculation including 10,000 optimization steps without structure shrinking to output the optimized structure of 640 kb resolution. In the same manner, we obtained the optimized structures of 320, 160, 80, 40, 20, and final 10 kb resolution. The calculations for the 100- and 1-kb resolution optimized structures are starting from the initial random structures of 1600 and 1024 kb resolutions, respectively.

**Assigning contact read to chromatin bead pair**. After dividing chromosomes into consecutive and equally sized beads, the contact read derived from the Hi-C experiment is assigned to the bead pair whose genomic positions correspond to those of the restriction fragment ends of the contact read. Therefore, we can obtain a contact map with its resolution corresponding to the size of the bead. Because a hierarchical protocol is employed, calculations are performed for a series of resolutions. For instance, to obtain the 10-kb optimized structure, calculations for 1280, 640, 320, 160, 80, 40, 20, and finally 10 kb resolutions were carried out. To efficiently map Hi-C contact reads into chromatin beads of various sizes, the Si-C approach firstly divides chromosomes into 1 kb beads, and then map all contact reads into such size beads. Second, the Si-C approach merges consecutive, 1-kb size beads into a specified size bead to obtain a new contact map of specified resolution and simultaneously accomplish the assignment of contacts in the new map. With the increase of the size of the bead, the case that the two ends of a contact share the same bead might occur. The contacts associated with such case are ignored during the structure calculations because they do not provide meaningful information on the structure.

**Reporting summary**. Further information on research design is available in the Nature Research Reporting Summary linked to this article.

## Data availability

Calculated 3D structures by Si-C are presented at https://github.com/TheMengLab/Si-C_3D_structure. 3D structures generated by Nucdynamics method in this study are presented at https://github.com/TheMengLab/Nuc_3D_structure_Cell1. Experimental data obtained from the published work has been summarized in Supplementary Table 1.

## Code availability

The source code of the Si-C method is available at https://github.com/TheMengLab/Si-C/tree/master/modeling. The minimum dataset for this code is deposited as GSM2219497_Cell_1_contact_pairs.txt in the repository https://github.com/TheMengLab/Si-C/tree/master/modeling. Codes for compartment identification and structural analysis are presented at https://github.com/TheMengLab/Si-C/tree/master/analysis. The DOI of the code is 10.5281/zenodo.4889467[34].

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

## Acknowledgements

We thank T. Stevens (University of Cambridge) and E. Laue (University of Cambridge) for answering our questions about NucDynamics. We also thank South China Normal University for the financial support. This work is also supported by the National Natural Science Fund of China NSFC 81502423, the Natural Science Foundation of Shanghai (20511101900,20ZR1427200), and SJTU Chen Xing Type B Project 16×100080032 (to Y.S.).

## Author contributions

L.M. conceived the study, developed the Si-C method, and devised the analyses. C.W. performed the calculations and participated in analyzing the data. Y.S. helped to interpret the experimental data downloaded from Gene Expression Omnibus (GEO) repository. L.M. supervised the work. L.M. and Q.L. drafted the manuscript with input from all the authors.

## Competing interests

The authors declare no competing interests.
