## [Peer Review File · Nature Communications]

REVIEWER COMMENTS

Reviewer #1 (Remarks to the Author):

The authors developed a Single-Cell Chromosome Conformation Calculator (Si-C) within the Bayesian theory framework and applied this approach to reconstruct intact genome 3D structures from the single-cell Hi-C data of eight G1-phase haploid mouse ES cells. To model 3D genome structure is a challenge already, let alone with single-cell data. I have the following concerns.

Major

1. Si-C need a more comprehensive assessment, compare to more 3D structure modeling tools in addition to NucDynamics. For example, ISD (DOI: 10.1371/journal.pcbi.1005292) , MBO (DOI: 10.1371/journal.pcbi.1004396) , ShRec3D (DOI: 10.1038/nmeth.3104) , SCL (doi: 10.1093/bioinformatics/btz181) and methods used in original single cell Hi-C papers (<https://doi.org/10.1038/nature23001> and doi: 10.1126/science.aat5641) .

2. The percentage of contact restraints may not be a proper index for the assessment.

a. Because the contact restraints depend a lot on the parameters used in the modeling methods. For example, the NucDynamics can generated a population of models with the full spectrum of contact restraints.

b. For a given contact number, higher resolution (small bin-size) will natural results smaller contact restraints. However, this does not necessary meaning a better model, as the confidential interval of higher resolution models are substantially lower than big binsize ones. Thus, the real meaning of Fig2a is questionable.

c. Why the violation percentage in Fig2a and 2b were not consistent?

3. Si-C performed significant worse with RMSD than NucDynamics.

4. A better assessment may be use reconstruction with sparse and noisy distance information (DOI: 10.1371/journal.pcbi.1004396) , or validation with 3D-FISH (doi: 10.1093/bioinformatics/btz181) .

5. The analysis and conclusion in the section "Conserved 3D whole-genome architecture in all cells" has already been discussed in the origin paper of NucDynamics.

Minor

1. It was commonly suggested a less than 2 RMSD median value of, what is the motivation behind the suggestion in the line 340 that RMSD should not more than 4. Moreover, evidence for the judgement that 4kb resolution is, rather than canonical 30kb, proper.

2. In Line360, the citation for harmonic oscillator potential is needed and explain way it is proper for the Bead-on-a-string representation.

3. In line382 and 391, the definitions do not satisfy $p(c_{ij} \geq 0 | r_{ij}) = 1$.

4. In Line408, the formula utilizes the continues of i and $i+1$, suggesting it applies to a single chromosome, not the whole genome. Thus, it is needed to explain how to apply this to the whole genome.

5. In Line408, an explanation needed for the unit coefficient on both $\epsilon_{cont}(r_{ij}, c_{ij})$ and harmonic oscillator potential.

6. What is the biologic meaning of the distribution of Fig.4b A/B compartment 'two shells and a core'? One may compare the results in fig4 with GPSeq(Girelli G. et al. Nat Biotechnol. 2020.), which experimentally assessed the such distribution.

Reviewer #2 (Remarks to the Author):

The authors describe a novel method, Si-C, to determine 3D chromatin structures from single cell Hi-C data. They compare their method to one existing tool, to itself across various resolutions and cell types, and to known features of 3D chromatin structure. Based on the data presented it is both faster and more accurate than the existing tool NucDynamics. The structures derived by Si-C agree well with known features of chromatin biology and seem consistent across cell types and resolutions. The paper is clearly written and well organized. I have some feedback below that might improve the manuscript, but in general, I recommend the manuscript for publication.

1) The authors state that "When we began this study, there was only one published method ... that has the ability to reconstruct intact genome 3D structures from single-cell Hi-C data of mammalian cells." This sentence implies that there are now other software packages for this purpose. I understand that new software is constantly being released and I don't think it is necessary for the authors to benchmark their software against all others. But I do think it is important for the authors to be more explicit about which other software exists. Please explicitly mention all other software for modeling 3D structures from single cell Hi-C data and briefly mention how they work.

2) The authors mention that NucDynamics exhibits high violation percentages for resolutions higher than 100Kb and then only show 100Kb data for NucDynamics throughout the rest of the paper. But I was consistently interested in seeing the higher resolution NucDynamics data in all figures. If its structures are inaccurate, I would expect to see that inaccuracy in all figures. I don't think we need to see all NucDynamics data at all resolutions in all figures. But seeing some further comparing to higher resolution NucDynamics data would be helpful.

3) In figure 3, the domains are drawn as boxes that span both sides of the diagonal (top left and bottom right). This obscures the data making it hard to see if the data does or does not support those domain calls. Please remove two sides of each box. That is, only annotate the domain on the top left half of the plot so we can clearly see the data on the bottom right.

4) DNA loops are one of the most interesting aspects of 3D chromatin structure. While it is not entirely necessary for this paper, I (and I suspect other readers) would be very interested in seeing how often loop anchors (detected from the bulk data) are in contact in single cells.

Thank two reviewers for their comments. The detailed responses are listed as follows:

Reviewer #1

Comments:

The authors developed a Single-Cell Chromosome Conformation Calculator (Si-C) within the Bayesian theory framework and applied this approach to reconstruct intact genome 3D structures from the single-cell Hi-C data of eight G1-phase haploid mouse ES cells. To model 3D genome structure is a challenge already, let alone with single-cell data. I have the following concerns.

Author Reply: Thanks very much for the positive comments and the kind suggestions.

Major

1. Si-C need a more comprehensive assessment, compare to more 3D structure modeling tools in addition to NucDynamics. For example, ISD (DOI:10.1371/journal.pcbi.1005292) , MBO (DOI:10.1371/journal.pcbi.1004396, ShRec3D (DOI:10.1038/nmeth.3104) , SCL (doi: 10.1093/bioinformatics/btz181) and methods used in original single cell Hi-C papers (<https://doi.org/10.1038/nature23001> and doi: 10.1126/science.aat5641) .

Author Reply: In addition to NucDynamics, we explicitly mentioned all other software, including ISD, ShRec3D, MBO, and SCL, and briefly mentioned how they work. The downloaded code of SCL can only construct 3D structure of individual chromosome rather than intact genome. We executed the SCL method to infer chromosome 1 of cell 1 at the 100-kb resolution, and found that the time consumed is nearly equal that for reconstructing intact genome structure of 100-kb resolution by Si-C. We failed to download the codes of MBO and ShRec3D because we have no permission to access the servers where they are stored.

In the revised manuscript, paragraphs are added in Page 3 line 62 and in Page 4 line 77 to briefly mention the ShRec3D, MBO and SCL methods. Moreover, a paragraph is added in Page 17 line 346 to compare the execution speed of SCL with Si-C.

2. The percentage of contact restraints may not be a proper index for the assessment.

a. Because the contact restraints depend a lot on the parameters used in the modeling methods. For example, the NucDynamics can generated a population of models with the full spectrum of contact restraints.

Author Reply: Thanks for the nice suggestion. The consistence of the models inferred by data-driven methods with the experimental contact constraints should be a requisite to assess the validations of such methods. The authors of the two published data-driven methods, NucDynamics (*Nature*, 544, 59, 2017) and ISD (*Plos Comput Biol*, 12, e1005292, 2016), also used the percentage of contact restraints as one of indexes for assessing the methods. In our manuscript, the percentage of contact restraints is only one of four indices for the assessment of the Si-C method.

b. For a given contact number, higher resolution (small bin-size) will natural results smaller contact restraints. However, this does not necessary meaning a better model, as the confidential interval of higher resolution models are substantially lower than big bin-size ones. Thus, the real meaning of Fig2a is questionable.

Author Reply: Thanks for the nice suggestion. We replaced the misleading statement “Fig. 2a illustrates that the violation percentage of Si-C decreases to nearly zero when the resolution is improved from 100-kb to 1-kb for the all eight cells.” by the statement “Moreover, the Si-C structures at 10-kb and 1-kb resolutions also show very low violation percentages. We further ran NucDynamics to generate the structure ensemble of 10-kb resolution for Cell 1 and, however, find

its average violation percentage higher than 70%. In terms of the consistence between models and the contact constraints, the Si-C models of 100-kb, 10-kb, and even 1-kb are reliable.” in the revised manuscript Page 6 line 149. We also deleted the Fig. 2b of the original manuscript.

c. Why the violation percentage in Fig2a and 2b were not consistent?

Author Reply: We downloaded the code of NucDynamicsa and directly executed the code to infer 3D structures. The data displayed in Fig. 2b of the original manuscript are derived from such structures. However, the data shown in Fig. 2a are based on the published NucDynamics structures. To clarify the inconsistent, we sent an email to one of the authors of the NucDynamics method, Dr. Tim J. Stevens. He kindly replied on our email and told us that the code we downloaded do not include the best-model selection process and full ambiguous protocol which are performed for generating the published NucDynamics structures. He mentioned that the next version of NucDynamics will include best-model selection process and full ambiguous protocol.

3. Si-C performed significant worse with RMSD than NucDynamics.

Author Reply: RMSD is one of four indexes for the assessment of the Si-C method. The RMSD values within the Si-C structure ensembles of Cell 1, 2 and 3 are comparable to those of the published NucDynamics structure ensembles. Although it seems that the RMSD values within the Si-C structure ensembles for Cell 6 are very high, the high correlation between the distance matrixes derived from the two conformations of a large RMSD (see Fig. 3e, 3f, and 3g) demonstrates that the two conformations are very similar in terms of chromatin folding. The flexibility of the two conformations results in the large RMSD. Our analysis shows that the Si-C method enables to produce consistent structures in terms of chromatin folding.

4. A better assessment may be use reconstruction with sparse and noisy distance information (DOI:10.1371/journal.pcbi.1004396) , or validation with 3D-FISH (doi:10.1093/bioinformatics/btz181) .

Author Reply: Thanks for the nice suggestion. In the revised manuscript, we used the 3D-FISH data reported by the suggested publication to assess the validation of structures inferred by Si-C. We found that the Si-C structures of 10-kb and 100-kb resolutions show high correlation with the FISH data, with Pearson correlation coefficients of 0.889 and 0.931, respectively. Moreover, the correlation of the published NucDynamics structure is 0.888.

In the revised manuscript, the paragraphs in Page 8 line 175 and Fig. 2b are added to display the results.

5. The analysis and conclusion in the section “Conserved 3D whole-genome architecture in all cells” has already been discussed in the origin paper of NucDynamics.

Author Reply: The section “Conserved 3D whole-genome architecture in all cells” mainly describe, rather than analyze, the features of 3D genome architecture displayed by the Si-C structures. As shown in Fig. 4, the features displayed by the Si-C structures are very similar to those by the NucDynamics structure. The similarity provides a support for the validation of Si-C.

Minor

1. It was commonly suggested a less than 2 RMSD median value of, what is the motivation behind the suggestion in the line 340 that RMSD should not more than 4. Moreover, evidence for the judgement that 4kb resolution is, rather than canonical 30kb, proper.

Author Reply: Thanks for the nice suggestion. The motivation behind of the suggestion of 4 bds originates from the distance threshold between a contacted bead pair. The distance threshold is set to 2 bds, and thus one bead of the contacted bead pair can move in the scope of a sphere of diameter 4 bds, relative to the other bead. The above sentences also have been added in the revised manuscript in Page 18 line 371.

2. In Line360, the citation for harmonic oscillator potential is needed and explain way it is proper for the Bead-on-a-string representation.

Author Reply: It is a reasonable assumption that the distances between sequentially connected beads are around an equilibrium distance. This situation is very similar to the sequentially connected particles in the polymer. In polymer chemistry, harmonic oscillator potential is widely used to describe the interactions between sequentially connected particles. Therefore, here, we also used the potential to describe the connectivity between sequentially adjacent beads.

3. In line382 and 391, the definitions do not satisfy $p(c_{ij} \geq 0 | r_{ij}) = 1$.

Author Reply: Technically, the definitions in line 381 and 392 of the original manuscript should be

$$P(c_{ij} \neq 0 | r_{ij}) = \begin{cases} g * \left(\frac{r_0 - r_{ij}}{r_0 + r_{ij}} \right)^{\frac{c_{ij}}{2}} & \text{for } c_{ij} \neq 0 \text{ and } r_{ij} < r_0 \\ 0 & \text{for } c_{ij} \neq 0 \text{ and } r_{ij} \geq r_0 \end{cases}$$

$$P(c_{ij} = 0 | r_{ij} < r_0) = g * \left(\frac{1}{1 + e^{-4(r_{ij} - r_0)}} \right)^{0.075} \quad \text{for } c_{ij} = 0$$

where g is the normalized factor that makes $p(c_{ij} \geq 0 | r_{ij})$ equal to 1. The value of g is independent with respect to the Hi-C data. Therefore, we ignored the contribution of g to the potential energy. In the revised manuscript, more rigorous definitions (in line 415 and 424) are used:

$$P(c_{ij} \neq 0 | r_{ij}) \propto \begin{cases} \left(\frac{r_0 - r_{ij}}{r_0 + r_{ij}} \right)^{\frac{c_{ij}}{2}} & \text{for } c_{ij} \neq 0 \text{ and } r_{ij} < r_0 \\ 0 & \text{for } c_{ij} \neq 0 \text{ and } r_{ij} \geq r_0 \end{cases}$$

$$P(c_{ij} = 0 | r_{ij} < r_0) \propto \left(\frac{1}{1 + e^{-4(r_{ij} - r_0)}} \right)^{0.075} \quad \text{for } c_{ij} = 0$$

4. In Line408, the formula utilizes the continues of i and $i+1$, suggesting it applies to a single chromosome, not the whole genome. Thus, it is needed to explain how to apply this to the whole genome.

Author Reply: In Page 19 Line 392 of the revised manuscript, we emphasized that the adjacent beads ($i, i+1$) are those in the same chromosome. The total potential energy (line 441 in the revised manuscript), mentioned by the reviewer, are redefined as:

$$E(\mathbf{R}) = \sum_{i, i+1 \text{ in the same chr.}} \varepsilon_{p/lys}(r_{i, i+1}) + \sum_{i < j} \varepsilon_{cont}(r_{ij}, c_{ij}) E(\mathbf{R})$$

5. In Line408, an explanation needed for the unit coefficient on both $\varepsilon_{cont}(r_{ij}, c_{ij})$ and harmonic oscillator potential.

Author Reply: Because the option of the unit of potential energy do not affect the relative energies of conformations, both potential energies have an arbitrary unit.

6. What is the biologic meaning of the distribution of Fig.4b A/B compartment ‘two shells and a core’ ? One may compare the results in fig4 with GPSeq(Girelli G. et al. Nat Biotechnol. 2020.), which experimentally assessed the such distribution.

Author Reply: Thanks for the nice suggestion. We compared the distributions of A/B compartments in the structures from GPSeq, Si-C, and NucDynamics. For convenience, the distributions are respectively displayed in the following figures of **a**, **b** and **c**. The figure of **a** is from the published paper (Girelli G. et al. Nat Biotechnol. 2020.). In the Si-C/NucDynamics structure, A compartments occupy the inner ring region which is sandwiched by the outer ring and core regions of B compartments, while in the GPSeq structure A compartments localize at the core region. This difference is because the GPSeq data is obtained based on data from a population of cells, instead from individual cell. The distribution of A/B compartments points out the locations of the active chromatin segments and inactive segments. The consistent ‘two shells and a core’ distributions in the Si-C and NucDynamic structures indicate that inactive chromatin can cluster in the core region of nuclear space in individual cell.

Reviewer #2 (Remarks to the Author):

The authors describe a novel method, Si-C, to determine 3D chromatin structures from single cell Hi-C data. They compare their method to one existing tool, to itself across various resolutions and cell types, and to known features of 3D chromatin structure. Based on the data presented it is both faster and more accurate than the existing tool NucDynamics. The structures derived by Si-C agree well with known features of chromatin biology and seem consistent across cell types and resolutions. The paper is clearly written and well organized. I have some feedback below that might improve the manuscript, but in general, I recommend the manuscript for publication.

Author Reply: Thanks very much for the positive comments and the kind suggestions.

1) The authors state that “When we began this study, there was only one published method ... that has the ability to reconstruct intact genome 3D structures from single-cell Hi-C data of mammalian cells.” This sentence implies that there are now other software packages for this purpose. I understand that new software is constantly being released and I don't think it is necessary for the authors to benchmark their software against all others. But I do think it is important for the authors to be more explicit about which other software exists. Please explicitly mention all other software for modeling 3D structured from single cell Hi-C data and briefly

mention how they work.

Author Reply: In addition to NucDynamics, we explicitly mentioned all other software for reconstructing 3D structures from single-cell Hi-C data in the revised manuscript, including ISD, ShRec3D, MBO, and SCL, and briefly mentioned how they work.

In the revised manuscript, paragraphs are added in Page 3 line 62 and in Page 4 line 77 to briefly mention the ShRec3D, MBO and SCL methods.

2) The authors mention that NucDynamics exhibits high violation percentages for resolutions higher than 100Kb and then only show 100Kb data for NucDynamics throughout the rest of the paper. But I was consistently interested in seeing the higher resolution NucDynamics data in all figures. If it's structures are inaccurate, I would expect to see that inaccuracy in all figures. I don't think we need to see all NucDynamics data at all resolutions in all figures. But seeing some further comparing to higher resolution NucDynamics data would be helpful.

Author Reply: Thanks for the nice suggestion. Both of Si-C and NucDynamics are data-driven methods, the consistence of their models with the experimental contact constraints should be a requisite to assess the validations of the inferred models. The 10-kb structure of NucDynamics shown in the following figure of b has a violation percentage higher than 70%. Therefore, we identified such structure as an inaccurate structure. However, it is hard to estimate whether the 10-kb structure of NucDynamic is accurate based on a quick glance at the inferred structure. For example, the following figures display the Si-C and NucDynamic calculated structures of cell 1 of 10-kb resolutions. It is difficult to distinguish the essential difference between the two based on a superficial comparison. Therefore, we do not think it is necessary to display higher resolution NucDynamics structures in the revised manuscript.

3) In figure 3, the domains are drawn as boxes that span both sides of the diagonal (top left and bottom right). This obscures the data making it hard to see if the data does or does not support those domain calls. Please remove two sides of each box. That is, only annotate the domain on the top left half of the plot so we can clearly see the data on the bottom right.

Author Reply: Thanks for the nice suggestion. The two sides of each box have been removed in Fig. 3 of the revised manuscript.

4) DNA loops are one of the most interesting aspects of 3D chromatin structure. While it is not entirely necessary for this paper, I (and I suspect other readers) would be very interested in seeing how often loop anchors (detected from the bulk data) are in contact in single cells.

Author Reply: Thanks for the nice suggestion. We mapped the 3331 CTCF/cohesion loops which are detected from the high-resolution Hi-C data of mouse Blymphoblastes onto the eight Si-C

10-kb structures, and found that less than 3% of the loops are formed in each individual cells and the percent obviously varies from cell to cell. The result are shown in the paragraph at Page 13 Line 264 and Supplementary Fig.1 of the revised manuscript.

REVIEWER COMMENTS

Reviewer #1 (Remarks to the Author):

The author added more comparison to other tools. However, from the comparison, I am not be convinced that the Si-C is a tool with sufficient novelty and outperformed to the exists tools to let it be published in Nature Communications.

Major:

In Fig 1d. the RMSDs were absolutely NOT comparable to NucDynamics in any cells. Even for cell 1,2, and 3. It is not convincing to me that the high RMSD was fully due to the polymer chain issue by just one example. Moreover, the RMSDs of NucDynamics were consistently, and substantially lower than Si-C in all cells, suggesting the polymer chain issue, if it does exist, shall be able to be avoided.

The authors reported only 3% of CTCF loops that found in high-resolution Hi-C were found in Si-C 10-kb data. Given the bulk Hi-C loops represent the high frequent contact events occurred in the cell population, one shall expect a not too low frequency of such events in individual cells. If the authors want to say 3% is a big enough number, their may need to show most of, if not all, contacts have higher than 3% in the cells will appear in the bulk Loops. 8 cells may be too small, but there are many large-scale single cell Hi-C data available now.

Fig5.a. The GRs are not always, actually, I shall say most of the GR peaks were not aligned to the boundaries of TAD-like structure. So, it is suspicious how the authors concluded that there exist highly extended chromatin chains. Indeed, the author argued that those internal GR peaks may represents the loops or hairpin-like structures. However, the clarification of this from an evidenced based conclusion to an ad hoc statement remains short. Moreover, such detailed structure, e.g., the 10kb showed in this study, inferred from ultrasparse single cell data need much external evidences to support.

Minor :

1. I found little evidence to support the "biologically valid" in the title.
2. Fig 1.b should also have the NucDynamics data included.
3. The language can be further polished.

Reviewer #2 (Remarks to the Author):

The authors addressed my concerns and I recommend for publication.

Thank two reviewers for their comments. The detailed responses are listed as follows:

Reviewer #1

Comments:

The author added more comparison to other tools. However, from the comparison, I am not be convinced that the Si-C is a tool with sufficient novelty and outperformed to the exists tools to let it be published in Nature Communications.

Author Reply: Modeling 3D genome structure from the sparse single-cell Hi-C data is a challenge and still in development. The existing tools include the following types: (1) MDS-based methods, including ShRec3D (*Nat Methods* **11**, 1141-1143 (2014)) and MBO (*Plos Comput. Biol.* **11**, e1004396 (2015)); (2) constraints optimization methods, including NucDynamics (*Nature* **544**, 59-64, (2017)) and SCL (*Bioinformatics* **35**, 3981-3988, (2019)); and (3) Bayesian-based method, including ISD (*Plos Comput. Biol.* **12**, e1005292 (2016)). The single-cell Hi-C data are sparse and do not determine the structure uniquely. However, the MDS-based methods and constraints optimization methods implicitly assume that there is a unique 3D structure underlying the sparse single-cell data and therefore lack a sound statistical foundation for inferring a 3D structure from the sparse single-cell Hi-C data. The Bayesian-based method, ISD, can infer statistical ensembles of chromosome structures from extremely sparse single-cell Hi-C data. However, the application of the ISD method to chromosome structure determination suffers from huge resource-consuming. According to the literature available, the NucDynamics method has successfully inferred intact 3D genome structure from single-cell Hi-C data of mammalian cells, while the other tools have only been applied to structure determinations of individual chromosomes.

The present Si-C method is a Bayesian-based method which combines the advantages of ISD and NucDynamics and enables rapid inference of statistically valid whole-genome structure ensemble at high resolution from single-cell Hi-C data of mammalian cell. We applied the Si-C approach to structure determinations from the single-cell Hi-C data of eight G1-phase haploid mouse embryonic stem (ES) cells. The validity of the Si-C inferred structures of the 100kb and 10kb resolutions are supported by the 3D-FISH experiments (*Nature* **543**, 519, (2017); *Science* **362**, 419 (2018)). Moreover, the average 3D structure derived from the Si-C inferred structures shows great correlation with the bulk Hi-C data (*Nature* **485**, 376-380 (2012)), providing a further support for the validity of the Si-C structures. Additionally, the execution speed of Si-C is

higher than that of NucDynamics and SCL. In a word, Si-C is an efficient Bayesian-based method for entire genome 3D structure determinations from single-cell Hi-C data of mammalian cells. Si-C outperforms to NucDynamics in terms of statistical foundation. Si-C outperforms to ISD in terms of resource-consuming. There is no doubt that Si-C outperforms the existing tools.

Major

1. In Fig 1d, the RMSDs were absolutely NOT comparable to NucDynamics in any cells. Even for cell 1,2, and 3. It is not convincing to me that the high RMSD was fully due to the polymer chain issue by just one example. Moreover, the RMSDs of NucDynamics were consistently, and substantially lower than Si-C in all cells, suggesting the polymer chain issue, if it does exist, shall be able to be avoided.

Author Reply: It is well known that the single-cell Hi-C data are very sparse. The authors of ISD (*Plos Comput. Biol.* **12**, e1005292 (2016)) pointed out that “Structure ensembles reconstructed from single-cell Hi-C contacts are typically less well defined than NMR structure ensembles. Therefore traditional measures to characterize and compare structure ensembles such as RMSDs are of limited used.” In addition to using RMSD, we also assess the conformation variability within structure ensemble through analyzing the distances matrices computed from replicas in the ensembles. Our analysis shows that the Si-C method enables to produce consistent structures in terms of chromatin folding.

On the other hand, it should be emphasized that the RMSDs of NucDynamics displayed in Fig. 1d are calculated from the published structures in (*Nature* **544**, 59-64, (2017)). We also directly ran NucDynamics using the same input Hi-data as that for Si-C to generate structure ensembles for all eight cells. Here, we refer to our calculated NucDynamics structures as the calculated NucDynamics structures in order to distinguish them from the published NucDynamics structures. The RMSDs of the calculated NucDynamics ensemble are displayed in yellow in the following Figure r1. Obviously, the Si-C RMSDs are comparable to the RMSD of the calculated NucDynamics ensemble for each cell. We discussed the inconsistent between the published and the calculated NucDynamics structures with one of the authors of NucDynamics, Dr. Tim J. Stevens. He told us that the NucDynamics code we downloaded do not include the best-model selection process or full ambiguous protocol which are performed for generating the published NucDynamics structures. He also mentioned that the next version of NucDynamics will include

best-model selection process and full ambiguous protocol. Here, we would like to thank Dr. Tim J. Stevens for his beneficial discussion.

As pointed out by the authors of ISD that RMSD should be limited used to characterize structure ensembles reconstructed from single-cell Hi-C contacts, we therefore, do not think it is necessary to display the Figure r1 in the revised manuscript.

Figure r1. Boxplot of pairwise RMSDs within the Si-C ensembles at 100-kb (purple), 10-kb (red), and 1-kb (green) resolutions along with pairwise RMSDs within the published NucDynamics ensembles (blue) [structures downloaded from GEO with accession code GSE80280] and the calculated NucDynamics ensembles (yellow).

2. The authors reported only 3% of CTCF loops that found in high-resolution Hi-C were found in Si-C 10-kb data. Given the bulk Hi-C loops represent the high frequent contact events occurred in the cell population, one shall expect a not too low frequency of such events in individual cells. If the authors want to say 3% is a big enough number, their may need to show most of, if not all, contacts have higher than 3% in the cells will appear in the bulk Loops. 8 cells may be too small, but there are many large-scale single cell Hi-C data available now.

Author Reply: We mapped the 3331 CTCF/cohesion loops which are detected from the high-resolution Hi-C data of mouse Blymphoblastes (*Cell* **162**, 687 (2015)) onto the eight published NucDynamics 100-kb structures, the eight Si-C 100-kb structures, and the eight Si-C 10-kb structures, respectively. As shown in Figure r2a, more than 50% of the loops are formed in the 100-kb inferred structures for both Si-C and NucDynamics, while the percentage decrease to less than 3% in the Si-C 10-kb structures (Figure r2b). However, as shown in Figure r2a, the percentage of loops that formed in the Si-C 10-kb structures is substantially comparable to that in

the 100-kb structures, when we use a 100 kb fragment (denoted as anchor fragment) to represent a loop anchor where the position is the midpoint bead of the anchor fragment, and define the contacted pair of loop anchors in the 10-kb structure with the minimum distance between the pair of anchor fragments (when the minimum distance is less than two bead diameters, the loop is regarded as to be formed). The similarity between the loop percentages of the Si-C 10-kb structures and the published NucDynamics structures provide a support for the reliability of the Si-C 10-kb structures.

We do not quite understand the meaning of the reviewer's opinion "If the authors want to say 3% is a big enough number, their may need to show most of, if not all, contacts have higher than 3% in the cells will appear in the bulk Loops." We guess the reviewer's opinion means that if a pair of loci has a probability higher than 3% to be identified as a contact in a population of cells, the pair of loci should appear as loop anchor in bulk Hi-C map. If we don't guess wrong, we do not agree with the reviewer's opinion. First, the 3% in our manuscript means the percentage of the 3331 CTCF/cohesion loops appear in one individual cell, instead of the probability for an individual loop appearing in populated cells. Second, as pointed out by Rao et al. (Cell 159, 1665 (2014)), the pairs of loop anchors correspond to pixels with higher contact frequency than typical pixels in their neighborhood. In other words, the loops in bulk Hi-C are not defined as whether the contact frequencies between the pairs of loci are larger than a threshold value or not. However, according to the reviewer's opinion, if there are evidences to prove that 3% is a big enough number, the number should be the threshold to define loops.

It can be seen that in Figure r2b (which is the Supplementary Fig. 2 in the revised manuscript), the percentage obviously varies from cell to cell, while such character becomes less obvious in Figure r2b. Moreover, there is no evidence to point out which percentage is reasonable. Therefore, we do not think it is necessary to display Figure r2a in the revised manuscript.

Figure r2: (a) Plot of the percentage of loops that are formed in the published NucDynamics 100-kb structures (black), the Si-C 100-kb structures (red), and the Si-C 10-kb structures with 100-kb anchor fragment (green). (b) Plot of the percentage of loops in the Si-C 10-kb structures.

3. Fig5.a. The GRs are not always, actually, I shall say most of the GR peaks were not aligned to the boundaries of TAD-like structure. So, it is suspicious how the authors concluded that there exist highly extended chromatin chains. Indeed, the author argued that those internal GR peaks may represent the loops or hairpin-like structures. However, the clarification of this from an evidenced based conclusion to an ad hoc statement remains short. Moreover, such detailed structure, e.g., the 10kb showed in this study, inferred from ultrasparse single cell data need much external evidences to support.

Author Reply: On examining the 3D structures shown in Fig. 3b, we can qualitatively consider that neighboring domains are linked by an extended chromatin chain. The positions of these linking chains correspond to the boundaries of domains in Fig. 3a, and each boundary align to a GR peaks which correspond to regions with higher extension than their neighborhood. Based on the 3D structures and the GRs of the boundary regions, we conclude that neighboring domains are linked by a highly extended chromatin chain. The distributions of GR shown in Fig. 3a demonstrate that domains are not uniformly compacted structures and there would be detail structures. In Fig. 3c and 3d, we show the loop and hairpin-like hyperfine structures which are embodied in a domain region to give a visual way to investigate domain structures. Also, we do not think these hyperfine structures must align to GR peaks.

We think some statement in the paragraph “Unparalleled details of chromatin folding provided by Si-C models” might cause misunderstanding. We are sorry for that. We have improved the statement in the revised manuscript.

Minor:

1. I found little evidence to support the “biologically valid” in the title.

Author Reply: The validity of the Si-C inferred structures are supported by the bulk Hi-C map and the observations of 3D FISH experiments. Therefore, we think the Si-C inferred structures are biologically valid. However, the key point of the present work is to introduce the Si-C method not to address the biological implication of the inferred 3D structures. So we delete the “biologically valid” in the title.

2. Fig 1.b should also have the NucDynamics data included.

Author Reply: Thanks for the nice suggestion. The following figure displays the Correlation between the 3D FISH results and the published NucDynamics 100-kb structures and the Si-C 100-kb structures. The following figure is shown in Supplementary Fig. 1 in the revised manuscript.

Figure 3r. Correlation between the median spatial distances measured by eight 3D FISH probe pairs (from ref. 22) and the median distances of the corresponding pairs in the Si-C 100-kb structures (a) and in the published NucDynamics 100-kb structures (b).

3. The language can be further polished.

Author Reply: Thanks for the nice suggestion. We have improved the language in the revised manuscript.

Reviewer #2

The authors addressed my concerns and I recommend for publication.

Author Reply: Thanks very much for the positive comment.

REVIEWERS' COMMENTS

Reviewer #1 (Remarks to the Author):

1. For the RMSD issue. The authors seems agree that Si-C performed not as good as the published NucDynamics, although their claimed that they know the authors of NucDynamics did not post the full/perfect version of the code. Even that is the case, the full/perfect code of NucDynamics is exists. So, I think it is graceful to clearly point out this affair in the revised manuscript.

Thank the reviewer for the comments. The detailed responses are listed as follows:

Reviewer #1

Comments:

1. For the RMSD issue. The authors seems agree that Si-C performed not as good as the published NucDynamics, although their claimed that they know the authors of NucDynamics did not post the full/perfect version of the code. Even that is the case, the full/perfect code of NucDynamics is exists. So, I think it is graceful to clearly point out this affair in the revised manuscript.

Author Reply: Thanks for the comment. In the revised manuscript, we add one paragraph(line 210-218) to discuss the ensemble RMSD for calculated genome structures using downloaded NucDynamics code and put the comparison of RMSD for structure ensembles generated by different methods in Supplementary Fig.1.